# MADS-Box Family Genes in *Lagerstroemia indica* and Their Involvement in Flower Development

**DOI:** 10.3390/plants13050709

**Published:** 2024-03-01

**Authors:** Zhongquan Qiao, Fuyuan Deng, Huijie Zeng, Xuelu Li, Liushu Lu, Yuxing Lei, Lu Li, Yi Chen, Jianjun Chen

**Affiliations:** 1Hunan Provincial Key Laboratory of Forest Clonal Breeding, Hunan Academy of Forestry, Changsha 410004, China; qzq110@hnlky.cn (Z.Q.); dfyyy22@163.com (F.D.); run507@163.com (H.Z.); 2College of Life Science and Technology, Central South University of Forestry and Technology, Changsha 410004, China; lxlukiyo@163.com (X.L.); 18878103452@163.com (L.L.); lyx1243954151@163.com (Y.L.); L942828878@163.com (L.L.); 3Mid-Florida Research and Education Center, Environmental Horticulture Department, University of Florida, 2725 S. Binion Road, Apopka, FL 32703, USA

**Keywords:** *Lagerstroemia indica*, MADS box, gene family analysis, flower organs, infertility, plant hormones

## Abstract

MADS-box is a key transcription factor regulating the transition to flowering and flower development. *Lagerstroemia indica* ‘Xiang Yun’ is a new cultivar of crape myrtle characterized by its non-fruiting nature. To study the molecular mechanism underlying the non-fruiting characteristics of ‘Xiang Yun’, 82 MADS-box genes were identified from the genome of *L. indica*. The physicochemical properties of these genes were examined using bioinformatics methods, and their expression as well as endogenous hormone levels at various stages of flower development were analyzed. The results showed that *LiMADS* genes were primarily classified into two types: type I and type II, with the majority being type II that contained an abundance of cis-acting elements in their promoters. By screening nine core proteins by predicted protein interactions and performing qRT-PCR analysis as well as in combination with transcriptome data, we found that the expression levels of most *MADS* genes involved in flower development were significantly lower in ‘Xiang Yun’ than in the wild type ‘Hong Ye’. Hormonal analysis indicated that ‘Xiang Yun’ had higher levels of iP, IPR, TZR, and zeatin during its early stages of flower development than ‘Hong Ye’, whereas the MeJA content was substantially lower at the late stage of flower development of ‘Hong Ye’. Finally, correlation analysis showed that JA, IAA, SA, and TZR were positively correlated with the expression levels of most type II genes. Based on these analyses, a working model for the non-fruiting ‘Xiang Yun’ was proposed. During the course of flower development, plant hormone response pathways may affect the expression of MADS genes, resulting in their low expression in flower development, which led to the abnormal development of the stamen and embryo sac and ultimately affected the fruiting process of ‘Xiang Yun’.

## 1. Introduction

Flowers serve as primary reproductive organs in angiosperms. A typical flower is composed of four concentric whorls of structures: the calyx, petals, stamens, and pistils. The flowering process in plants is governed by a multitude of signals and a sophisticated regulatory network involving numerous genes [1]. Currently, the prevailing molecular mechanism of plant flowering is based on the ABCDE model proposed for *Arabidopsis thaliana* and *Antirrhinum majus* [2]. These five classes of genes exhibit distinct homozygous and heterozygous functions, which are associated with various developmental processes. Specifically, genes in classes A and E operate in whorl I to establish sepals, while genes in classes A, B, and E participate in whorl II to determine petals. In whorl III, genes in classes B, C, and E contribute to the formation of the stamens, whereas genes in classes C and E are responsible for the development of the carpels in whorl IV. Lastly, genes in classes C, D, and E are involved in specifying ovules [3,4].

In the ABCDE model, all genes, excluding *AP2*, belong to the MADS box gene family [5]. The MADS box gene family is comprised of transcriptional regulators with vital functions and encompasses a broad group of gene families. They are extensively distributed throughout the plant kingdom and play important regulatory roles in plant growth, development, and signaling processes, such as vernalization, inflorescence meristem differentiation, embryo development, and fruit ripening [6,7]. Based on the C-terminal structure, functional role, and evolutionary rate of MADS-box proteins, the MADS-box family can be categorized into two distinct taxa: Type I and Type II [8]. Type I genes consist solely of the MADS structural domain and do not contain the K domain. These can be further divided into three subfamilies: Mα, Mβ, and Mγ. Research has demonstrated that MADS-box genes are involved in the regulation of female gametophyte and endosperm development in *Arabidopsis* and Gramineae [9]. Type II genes feature three more structural domains than Type I genes: a more conserved intervening structural domain (I structural domain), a moderately conserved keratin-like helical structural domain (K structural domain), and a variable C-terminal structural domain (C structural domain) [10,11]. The MADS box genes can be divided into two subfamilies, MIKC* and MIKC^C^. Research has demonstrated that *MIKC^C^*-type genes play a regulatory role in various aspects of sporophytic development in higher plants [12], while *MIKC**-type genes are essential for male gametophyte development in the model plant *A. thaliana* [13,14].

Plant hormones, such as auxin, gibberellin (GA), cytokinin (CK), and abscisic acid (ABA), play crucial roles in the regulation of various cellular activities, including cell division, differentiation, reproduction, and responses to abiotic and biotic stresses. For instance, IAA as auxin controls the differentiation of the meristem into vascular tissue and promotes the development of various organs [15]. GAs, on the other hand, are mainly responsible for seed dormancy, bud elongation, seed germination, fruit, and flower maturation. For example, GA induces seed germination when exposed to cold or light [16]. ABA often interacts with other hormones, such as GA and cytokinin, to inhibit seed germination and post-germination growth, protecting plants from abiotic stresses [17]. CKs can maintain the growth potential of the stem tip meristem (pluripotency), promoting cell division and increasing cell expansion during the proliferative and amplified phases of leaf cell development [18]. Furthermore, other hormones, including jasmonic acid (JA) and salicylic acid (SA) play significant roles in plant responses to external stresses.

*Lagerstroemia indica*, a woody ornamental plant belonging to the family Lythraceae [19,20], is renowned for its stunning, colorful blossoms, extended flowering period, and resistance to air pollution. These features make it an ideal choice for planting in courtyards, parks, roadside areas, indoor bonsai, and cut-flower arrangements. It is a highly regarded summer ornamental tree species in China. By chance, our research team discovered a mutant of *L. indica*, which was named ‘Xiang Yun’ [21]. This cultivar is devoid of fruit-bearing capabilities and has a longer flowering period than the typical *L. indica*. Early observations of its phenotypic and physiological indicators revealed that the pollen was inactive and microspore fertility was low. The anatomic study on the process of differentiation showed that the embryo sac abortion was caused by the abnormal tapetum layer during the embryo sac development of ‘Xiang Yun’ [22]. However, the underlying mechanism by which genes are responsible for its flower organ development and interactions with plant hormones remains unclear.

In this study, we utilized bioinformatics techniques to identify the MADS gene family members in the whole genome of *L. indica* and analyzed their physicochemical properties, structural characteristics, and cis-acting elements. Through the comparison of differences between ‘Xiang Yun’ and wild type ‘Hong Ye’ in gene expression and endogenous hormone levels during flower organ development, a working model underlying the lack of fruit setting was proposed. The results from this study could provide a theoretical basis for understanding the developmental mechanisms of *L. indica* flower organs and for exploring the molecular basis of sterility observed in ‘Xiang Yun’.

## 2. Results

### 2.1. Identification of LiMADS Gene Family Members and Physicochemical Analysis

A comprehensive screening of the MADS candidate gene family in *L. indica* was conducted using the resources provided by NCBI and SMART. Following the elimination of inaccurate genes, 82 *L. indica MADS* genes were identified, and their physiochemical properties were analyzed. Each gene was subsequently named and numbered according to its respective chromosomal location and designated as *LiMADS1*–*LiMADS82* (Table 1 and Appendix A). The molecular weight and amino acid length of the MADS gene family varied. LiMADS53 is the shortest at 8.74 kDa and 75 residues, and LiMADS50 is the longest at 199.61 kDa and 1784 residues. The physicochemical properties of MADS proteins differed considerably. The isoelectric points ranged from 4.54 to 10.31, whereas the instability indices indicated that the majority of the MADS proteins were unstable, with values ranging from 34.42 to 80.52. Furthermore, the fat coefficients exhibited a broad distribution, ranging from 54.21 to 108.30. Interestingly, the hydrophilicity indices were consistent with those of the hydrophilic proteins, except for MADS22. These findings suggest that MADS proteins possess a wide range of physicochemical properties. Results from subcellular localization prediction indicated that the MADS proteins were all localized in the nucleus, which was consistent with the characteristics of transcription factors, indicating that they all functioned in the nucleus.

### 2.2. Phylogenetic Analysis of LiMADS Gene Family Members

To elucidate the evolutionary relationship of *LiMADS* family members in *L. indica*, we conducted multiple sequence comparisons and clustering analyses of 82 MADS proteins isolated from *L. indica* with 107 MADS proteins from *A. thaliana*. Furthermore, their phylogenetic trees were constructed (Figure 1). Results showed that LiMADS-box members could be divided into two distinct groups, type I and type II, based on the classification of MADS-box proteins in *A. thaliana*. We identified three subfamilies within type I proteins, Mα, Mβ, and Mγ, based on their structural properties. In *L. indica*, Mα consisted of 17 out of 28 type I proteins, whereas Mβ and Mγ included 2 and 9 proteins, respectively (Figure 1a). Notably, the Mα proteins in the Mα group, which were homologous to AGL23 and AGL61 as well as to the *L. indica* LiMADS9 and LiMADS61, might play a role in the development of female gametophytes and embryos [23,24]. The proposed function of LiMADS25 and LiMADS62 in the Mβ subgroup was to regulate the nutrient supply to the outer testa and female gametophytes. Conversely, homologous genes of *L. indica* in the Mγ subgroup, such as AGL80, LiMADS14, and LiMADS28, could be responsible for promoting endosperm development with endosperm-specific functions [25,26]. The Type II proteins were characterized by the presence of MIKC^C^ and MIKC*. Out of these Type II proteins, 49 belonged to MIKC^C^, while only 5 belonged to MIKC* (Figure 1b). MIKC^C^ is a vital regulator of floral organ differentiation and was further classified into 14 subgroups based on its evolutionary history. These subgroups included FLC, AG, AGAMOUS, AGL6, GGM13, SQUA, STMADS11, DEF, and GLO, which could be important for regulating floral organ development. Additionally, subgroups, such as AGL12 [27], AGL17 [28], and TM3 [29] were involved in regulating root development, while others, such as AGL15 [30] were expressed predominantly during embryogenesis and seed development. Finally, there were specific subgroups responsible for stimulating flowering in the tips of the stems and axillary meristems. The AGL71 subpopulation [31] is crucial for the development of male gametophytes (pollen) in *A. thaliana*. The MIKC* complex plays a vital role in the regulation of pollen development by suppressing immature pollen genes and activating mature pollen genes. Structural domain transcription factors of the MIKC* complex are also essential [14]. Several genes expressed in the pollen of *L. indica*, including *LiMADS11*, *LiMADS46*, *LiMADS53*, *LiMADS57*, and *LiMADS75*, showed significant homology to *AGL30*, *AGL65*, *AGL66*, *AGL67*, *AGL94*, and *AGL104*.

### 2.3. Chromosome Distribution and Gene Collinearity Analysis of LiMADS in L. indica

Employing TBtools v2.052 software, we conducted a chromosome localization analysis using the annotation data of the *L. indica* genome. Results showed that 82 genes were unevenly distributed across 24 chromosomes, suggesting that MADS transcription factors played multiple roles in *L. indica*. These genes were designated *LiMADS1* to *LiMADS82* according to their chromosomal locations (Figure 2). Notably, no *MADS* genes were present on chromosomes 10 and 21, whereas chromosome 9 harbored the highest number of *MADS* genes (nine genes).

To examine the evolutionary relationships among the *LiMADS* genes in *L. indica*, we performed a BLAST comparison of the sequence similarity of all *LiMADS* genes (Figure 3a). The analysis revealed 29 pairs of large segment duplications and 9 pairs of closely linked genes among the 82 *LiMADS* genes, suggesting that tandem duplication of genes occurred. Interestingly, our analysis indicated that *LiMADS6*, *LiMADS7*, *LiMADS18*, *LiMADS19*, *LiMADS44*, and *LiMADS45* were closely linked and could play a significant role in driving gene evolution. Moreover, covariance analysis, employed to examine the covariance between the *LiMADS* gene family of *L. indica* and the *Arabidopsis* genome, enabled the clarification of the origin of *LiMADS* genes and their evolutionary relationships (Figure 3b). In this study, we identified 45 homologous *MADS* genes in *L. indica* and *A. thaliana*. The *MADS* homologous genes were predominantly located on chromosome 22 of *L. indica*, whereas chromosomes 1, 10, 11, 15, and 21 did not exhibit any *MADS* homologues. Moreover, multiple *AtMADS* homologues were observed for certain *LiMADS* genes, and each *AtMADS* gene had numerous *LiMADS* homologues.

The Ka/Ks ratio, which measures the rates of non-synonymous (Ka) and synonymous (Ks) substitutions in genetics, is a pivotal element in the evolution of nucleic acids [32]. This study demonstrated that gene pairs exhibited Ka/Ks values below 1, indicating evolution under purifying selection.

### 2.4. Analysis of LiMADS Gene Structure and Conserved Motifs in L. indica

Through the integration of phylogenetic trees, gene structures and motifs, we uncovered similarities and disparities in gene structures and evolutionary relationships between members of the *LiMADS* gene family. Our findings suggested that *L. indica* possesses 10 conserved motifs, with the number of conserved motifs in *L. indica* MADS-box proteins ranging from 2 to 12 (Figure 4b). All LiMADS proteins, comprising motif 1 and 2, comprised the typical MADS structural domains, encompassing approximately 70 amino acids. The Mα subclade of type I MADS-TF genes exhibited a predominance of motifs 5 and 8, while the Mβ subclade, comprising LiMADS25 and LiMADS62, was distinguished by the presence of only two motifs, with motifs 3 and 6 being exclusive to the Mγ subclade. The type II MADS-TF family displayed a conserved four-domain structure comprising the M, I, K, and C domains [32], with the K domain being a distinctive feature of MIKC plants and being highly conserved throughout evolution. Notably, the LiMADS11, LiMADS46, LiMADS53, LiMADS57, and LiMADS75 subfamilies within the MIKC* clade did not contain the K structural domain. The C-domain comprises various motifs that enable proteins to perform different functions (Figure 4d). On the C-terminus of the MIKC subfamily in *L. indica*, we identified several distinct motifs. By comparison to other plant species (*Punica granatum*, *Citrus sinensis*, and *Arabidopsis*), both LiMADS64 and LiMADS81 contain the FUL motif. LiMADS12 is homologous to *Arabidopsis* AGL24, and both contain the SVP motif in the C-terminal. LiMADS3, LiMADS10, and LiMADS24 belong to the AG subfamily and contain two AG motifs in the C-terminal. LiMADS37’s C-terminal contains a PI-derived motif with the sequence FxFRLQPSQPNLH and an euAP3 motif. The FUL, SVP, AG, and AP3 motifs are highly conserved across different species.

Our analysis of the gene structure suggested that most type I genes lacked introns, with only eight genes (*LiMADS14*, *LiMADS28*, *LiMADS17*, *LiMADS18*, *LiMADS70*, *LiMADS42*, *LiMADS20*, and *LiMADS60*) containing a small number of introns (Figure 4c). The results of our gene structure analysis indicated that a large proportion of type I genes lacked introns, with only eight genes (*LiMADS14*, *LiMADS28*, *LiMADS17*, *LiMADS18*, *LiMADS70*, *LiMADS42*, *LiMADS20*, and *LiMADS60*) containing a small number of them. On the other hand, all the type II genes possessed a substantial number of introns, except for four genes (*LiMADS5*, *LiMADS22*, *LiMADS53*, and *LiMADS80*) that did not have any introns. The average count of introns in the type II genes was significantly greater than that in the type I genes. *LiMADS30* had the longest intron, and *LiMADS2*, *LiMADS11*, and *LiMADS26* had longer introns. These genes were involved in floral transitions and had structural domains of expression typically identified in trophic tissues. Furthermore, this study revealed that intron length varied greatly among genes, indicating that type II genes have undergone increased genomic complexity during evolution.

We further cloned the *MADS* gene from both wild type *L. indica* ‘Hong Ye’ and mutant ‘Xiang Yun’ and compared their sequences (Appendix A). The results revealed that ‘Xiang Yun’ had a larger number of single nucleotide polymorphisms and base mutations compared with ‘Hong Ye’. The mutations were mainly transitions, with a few transversions, resulting in silent mutations and missense mutations in the gene sequence, which may affect the function of the protein.

### 2.5. Prediction of Cis-Acting Elements in the LiMADS Gene Family of L. indica

The promoter region of the *LiMADS* gene family of *L. indica* contains numerous cis-elements, which can be grouped into four categories: 25 light-responsive elements, 12 elements related to growth and development, 12 elements related to plant hormone response, and 6 elements related to biological and abiotic stresses. Among these, photoresponsive elements are the most diverse and abundant, with G-Box being present in most MADS genes (Figure 5). Light conditions may affect the expression of *LiMADS*, as suggested by the presence of hormone-responsive elements in *LiMDAS* promoters. These elements include growth factors such as AuxRE, AuxRR core, TGA elements, and TGA-box, as well as gibberellin (P-box, TATC-box, GAREmotif), ABRE, jasmonic acid (CGTCA motif, TGACG motif), and salicylic acid (SARE, TCA motif), mostly composed of abscisic acid and jasmonic acid response elements. *LiMADS* promoters contain stress response elements, including anaerobic induction elements (ARE, GC-motif), low temperature response elements (LTR), and drought response elements (MBS). Additionally, elements related to flower development, such as POLLEN1LAT52, GTGANTG10, and CArG, have been identified, mainly involved in pollen development (Appendix A). The *LiMADS* gene family was anticipated to exert a substantial influence on the growth, development, and response of *L. indica* to external environmental factors.

Furthermore, upon analyzing various subfamilies, it was found that type II genes had a higher number and more variety of promoters than type I genes (Appendix A). Notably, the MIKC^C^ subfamily contained a higher proportion of hormone and photoresponsive elements. Specifically, the AGL6, SQUA, AG, FLC, DEF, and GLO subfamilies exhibited a larger number of photoresponsive elements, whereas AGL15 and AGL17 had fewer. This finding may be related to their respective functions. Most of the genes are believed to be involved in hormonal pathways, as the Mα subfamily contains a smaller number of hormone elements compared to other subfamilies that have more cis-acting elements related to hormones.

### 2.6. Analysis of Protein-Protein Interaction Network of LiMADS Gene

To gain information regarding the interactions among LiMADS proteins, we employed a computational approach to predict the LiMADS protein-protein interaction network using homologous MADS-box proteins from *A. thaliana*. This analysis was expected to enhance our understanding of LiMADS protein interactions. The study found that nine LiMADS proteins (LiMADS7, 28, 29, 31, 37, 62, 71, 80, and 81) play a central role in the protein interaction network (Figure 6). Notably, LiMADS28 and LiMADS62 are type I genes and exhibit strong interaction with LiMADS31. LiMADS37, a class B gene similar to the AP3 gene in *A. thaliana*, is a transcription factor essential for normal petal and stamen development. LiMADS31 forms a heterodimer with PI (LiMADS51), a gene responsible for regulating flower development. Together with LiMADS71 and LiMADS63, LiMADS31 recognizes the floral meristem and regulates the expression of class B, C, and E genes. LiMADS80 and LiMADS81 are class E genes that are active in flowers and ovules. They interact with multiple classes of genes to form complexes that ensure the normal development of petals, stamens, and carpels.

### 2.7. Analysis of LiMADS Gene Expression Levels at Different Flower Development Stages

To investigate the regulation of flower organ development by the *LiMADS* gene family in the cultivar Xiang Yun, we conducted transcriptome sequencing on flower buds of ‘Xiang Yun’ and ‘Hong Ye’ at different stages of development (Figure 7). Our results indicated that the expression levels of *AGL6*, *AG15*, *ALG15*, *AGL17*, *DEF*, *GLO*, and *STMADS11* subfamily genes, which were involved in flower development, were higher in ‘Hong Ye’ and lower in ‘Xiang Yun’ (Figure 7). Notably, *AGL6*, *DEF*, *GLO*, *ALG15*, *AGL17*, *STMADS11*, and *MIKC** were related to pollen development in *A. thaliana* [12], and these genes were specifically highly expressed in ‘Hong Ye’. However, the expression level of ‘Xiang Yun’ was lower than the control variety ‘Hong Ye’ (Figure 7). This difference in expression was hypothesized to be associated with the development of stamens in ‘Xiang Yun’. The *TM3* subfamily of genes was known to regulate the expression of egg cells and embryos during the development of flower organs [28]. The heat map results indicated that the expression of *TM3* increased with the development of flowers in ‘Hong Ye’ but decreased in ‘Xiang Yun’. Therefore, it was speculated that the low expression of *TM3* in ‘Xiang Yun’ was indicative of a reduction in the development of egg cells and embryos, leading to the flowering but not the fruiting phenotype observed in ‘Xiang Yun’.

Previous research revealed that the variety ‘Xiang Yun’ possessed a more protracted flowering cycle than ‘Hong Ye’. A heat map depicting high levels of *FLC* gene expression in ‘Xiang Yun’ in relation to red leaves, and strong initial expression during the development of flower buds aligned with the function of *FLC* as a flowering inhibitor in *A. thaliana* (Figure 7). Whittaker [33] emphasized the importance of extended exposure to cold conditions in developing flowering ability, underscoring the crucial role of *FLC* as a determinant. This information supported the extended period of blooming observed in ‘Xiang Yun’.

Furthermore, based on the results of protein interactions, we selected nine core MADS proteins to determine the gene expression levels of *LiMADS* in ‘Hong Ye’ and ‘Xiang Yun’ (Figure 8). The qRT-PCR expression pattern of most *LiMADS* genes was consistent with the general trend of transcriptome expression patterns. During flower development, the expression levels of *LiMADS7*, *LiMADS29*, *LiMADS31*, and *LiMADS37* in ‘Hong Ye’ significantly increased. Additionally, there were some differences in expression levels between two cultivars during the same period. *LiMADS28*, *LiMADS62*, *LiMADS80*, and *LiMADS81* showed low levels of expression in ‘Xiang Yun’. However, the expression levels of *LiMADS71* were significantly higher in ‘Xiang Yun’ than in ‘Hong Ye’. This result differs from the transcriptome data, which may be due to differences in sequencing and qRT-PCR samples.

### 2.8. Changes in Hormone Levels during Organ Development of L. indica Flowers

To further investigate the mechanism of sterility in ‘Xiang Yun’, we utilized ‘Hong Ye’ as a control (Figure 9) and employed HPLC to assess 15 endogenous hormones during flower development. Our findings revealed substantial disparities among hormones. The levels of IAA and IBA increase with flower development. Specifically, in the cultivar ‘Xiang Yun’, the levels of IAA and IBA increased by 1.3 and 1.75 times, respectively, during the S5 period compared with the S1 period. Similarly, the levels of IAA and IBA in red leaves increased by 0.8 and 1.54 times, respectively. In contrast, ABA and SA levels decreased with the progression of flower development. The ABA content in ‘Xiang Yun’ decreased by 77.9% and 73.8% in S5 and S1 periods, respectively, while the SA content decreased by 58.8% and 73.9% in ‘Xiang Yun’ and ‘Hong Ye’, respectively.

Biological activities of GAs are typically limited to GA_1_, GA_3_, GA_4_, and GA_7_ in higher plants [34], and they play a critical role in regulating various aspects of plant growth and development, including seed germination, seedling morphogenesis, and maturation of different organs. GA_1_ and GA_7_ contents generally increased progressively through the development of the flower organs (Figure 10). GA_3_ and GA_4_ contents fluctuated in the developmental stages. Notably, the levels of GA_3_ (S1 and S2 stages) and GA_4_ (S1, S4, and S5 stages) were significantly higher in ‘Xiang Yun’ than ‘Hong Ye’ (Figure 10). JA is a key regulatory molecule in plant growth and development as well as in response to abiotic stress. Mutants with impaired JA synthesis or signaling during flower organ development exhibit delayed stamen growth and anther unclefting [35]. In this study, we found that JA content in ‘Xiang Yun’ was significantly lower than in ‘Hong Ye’. Moreover, the content of MeJA in ‘Xiang Yun’ did not change significantly with flower development, whereas the contents of S4 and S5 in ‘Hong Ye’ were 13.4 and 12.4 times higher than in ‘Xiang Yun’, respectively (Figure 10). This suggested that JA synthesis may be affected in ‘Xiang Yun’, which could be related to its sterility. IAA and IBA contents increased with the progress of flower development. IAA levels at S4 and S5 of ‘Xiang Yun’ were significantly higher than those of ‘Hong Ye’, but IBA contents at S3–S5 of ‘Xiang Yun’ were substantially lower than those of ‘Hong Ye’. Additionally, the levels of iP, IPR, TZR, and Zeatin in ‘Xiang Yun’ were significantly greater than those in ‘Hong Ye’ from S1 to S3. Specifically, the TZR content was 4.29, 5.77, and 3.3 times higher in ‘Xiang Yun’ than in ‘Hong Ye’ at the S2, S3, and S4 stages, respectively. SA and ABA exhibited a decreased trend with the development of flowers, MeSA remained largely stable, and their levels varied in both ‘Xiang Yun’ and ‘Hong Ye’.

Finally, we analyzed the correlation between hormone concentrations and gene expression levels in the transcriptome (Figure 11). The results showed that JA, IAA, SA, and TZR were positively correlated with the expression levels of most type II genes. Additionally, genes in *AGL6*, *AG*, and *TM3* were positively correlated with GA_3_, GA_4_, and GA_7_. On the other hand, ABA was negatively correlated with most genes.

## 3. Discussion

MADS-box genes are of paramount importance in various plant activities, including the formation of flowers, regulation of flowering time, and flower development [36]. As a result of the completion of genome sequencing in multiple species, there has been an increased understanding of the role of MADS-box genes in plant development. However, research on the function of the MADS gene family in *L. indica* is limited. In this study, we analyzed the MADS gene family in *L. indica* and identified 82 *LiMADS* genes, the number is similar to that reported by Zhao et al. [37] in pomegranate. A comparative phylogenetic analysis of *L. indica* and *Arabidopsis* resulted in the classification of *L. indica* into five subfamilies, with the *MIKC^C^* subfamily comprising 60% of the genes and exhibiting the most homologous genes compared to *Arabidopsis*. This finding suggested that the *MIKC^C^* subfamily was conserved across species. The *MIKC^C^* gene class is crucial for regulating floral organ characteristics and plays a vital role in the evolution of diploid sporophytic reproductive structures in seed plants [11]. The data indicated that only two Mβ genes were present in *L. indica*, which were significantly fewer in number than in *Arabidopsis*. This suggested that the Mβ gene in *L. indica* could be lost during evolution. Similar observations were reported in *Camellia sinensis* [38]. Consequently, the *MIKC^C^* gene underwent greater proliferation and diversification in *L. indica* than other *LiMADS* genes, potentially enabling it to perform multiple functions.

The function of a gene is dictated by its structure within the organism. The greater the number and length of introns in a gene sequence, the more diverse are the methods by which genes can be spliced, which subsequently affects gene expression and protein activity [39]. The intron average of type II genes in the MADS gene structure was significantly higher than that of the type I genes. This is similar to the MADS gene family in orchids [40], grapes [41], walnuts [42], and other plants, leading us to hypothesize that type II genes could play a critical role in the development of *L. indica* flower organs. Examination of cis-acting components within the *L. indica* MADS gene family revealed the presence of cis-acting elements associated with hormonal responses and abiotic stresses, specifically the methyl jasmonate response element, gibberellin response elements, light response elements, and flower development-related elements such as CArG. Comparable cis-acting elements have been identified in maize and lettuce [43,44], suggesting that the regulatory promoter elements of MADS-box genes are conserved across species. Consequently, the findings indicate that the MADS-box can respond to hormonal regulation. The hormones in question are of paramount importance for plant growth and development as well as in the response of plants to stress. Furthermore, the MADS-box family protein network was analyzed, and it was found that most MADS-box members interact with each other. LiMADS28 (AGL80) plays a crucial role in the protein interaction network and exhibits strong interaction with LiMADS19 (AGL61). In *Arabidopsis*, AGL80 and AGL61 are expressed in central cells through the formation of heterogenic complexes [24]. The Class B transcription factor LiMADS37 (AP3) consistently forms heterodimers with LiMADS51 (PI). The AP3/PI complex can also form ternary complexes with the class E transcription factor LiMADS80 (SEP3) to contribute to the development of petals and stamens [45]. Additionally, gene duplication is a critical mechanism in the development and evolution of gene families in plants [46]. Duplicated genes can evolve novel functions and enhance the capacity of plants to adapt to their environments [47]. In *L. indica*, tandem duplications were identified in nine gene pairs. The calculation of Ka and Ks values indicated that *LiMADS* genes were subjected to purifying selection, suggesting that their structure has contributed to the expansion of the *L. indica* MADS gene family, which plays a crucial role in increasing gene function diversity.

The primary causes of plant failure to reproduce have been ascribed to male or female sterility, and extensive research has been conducted on fertility genes and their associated mechanisms in model plants. To investigate the sterility mechanism of ‘Xiang Yun’, we analyzed gene expression levels using transcriptome data and qPCR. The results indicate that *LiMADS71* gene expression is high in ‘Xiang Yun’. We also found that *LiMADS71* may be homologous to SVP in *A. thaliana*, which can inhibit class B and class C flower homologous genes [48]. Further analysis revealed that genes responsible for flower meristem development (*LiMADS29*, and *LiMADS31*), stamen development (*LiMADS37*, *LiMADS62*, *LiMADS80*, and *LiMADS81*), and pistil development (*LiMADS28*) were expressed at lower levels in ‘Xiang Yun’. This suggests that the sterility of ‘Xiang Yun’ may be linked to the underdevelopment of its reproductive organs. Male sterility, which encompasses anther morphological defects, microsporogenesis issues, impaired pollen development, and disrupted pollen function, may also be a contributing factor [49]. Similarly, female sterility can arise from defects in the ovaries and ovules [50,51]. A previous study exploring the morphology of *L. indica* identified anther dehiscence and abnormal embryonic sac development as the primary causes of *L. indica* sterility. Furthermore, it is often observed that senescence-triggering hormones originate from the maturation of plants, and a deficiency in these hormones can extend their lifespan [52]. *FLC*, a suppressor of flowering, is significantly expressed during the early stages of flower bud development in ‘Xiang Yun’. This finding lends credence to the idea that ‘Xiang Yun’ has a longer flowering period compared to its ‘Hong Ye’ counterpart.

During flower development in angiosperms, sepal opening, petal growth, stamen development, style elongation, anther dehiscence (which releases pollen), and stigma maturation (which facilitates pollen germination and pollen tube growth) are all regulated by endogenous hormones [53]. MeJA levels were significantly higher in ‘Hong Ye’ than ‘Xiang Yun’. Furthermore, low levels of MeJA were detected in male sterile cotton [54], rice [55], and tomato [56], where the application of exogenous MeJA to early buds reversed anther sterility. Moreover, our correlation analysis results showed that JA was positively correlated with MADS gene expression. Therefore, we hypothesized that the synthesis of jasmonic acid in ‘Xiang Yun’ was associated with the development of stamens, which ultimately resulted in flowering failure. JA is known to co-regulate various developmental processes in plants with other plant hormones, such as cytokinin [57]. Our findings revealed that TZR, zeatin, and iP, were lower in ‘Hong Ye’ than in ‘Xiang Yun’. This may be attributed to the antagonistic relationship between JA and TZR, Zeatin, and iP during the development of flower organs, where high levels of JA inhibit TZR, Zeatin, and iP production.

In summary, we proposed a working model to illustrate how the MADS-box family genes interact with hormones to regulate the development of flower organs in *L. indica* ‘Xiang Yun’ (Figure 12). Specifically, plant hormone response pathways affect the expression of MADS genes, resulting in their low expression during the course of flower development. The reduced expression may lead to the abnormal development of stamen and embryo sac and ultimately hamper the fruiting process of ‘Xiangyun’. Obviously, the regulation mechanism underlying the sterility of ‘Xiang Yun’ is complex. This study only discusses genes in the MADS-box family and their interactions with hormones in flower development. It is known that flower development also depends on the regulation of proteins, such as LFY and UFO, and their involvement in the regulation of ‘Xiang Yun’ has not been studied. Thus, further research is warranted to verify the function of the MADS gene in *L. indica* flowering and specific mechanism underlying the sterility of ‘Xiang Yun’.

## 4. Materials and Methods

### 4.1. Identification of MADS Gene in the Genome of L. indica

To identify members of the MADS family within the unpublished genome of *L. indica*, we obtained a conserved domain Hidden Markov Model for MADS (PF00319) from the Pfam protein database. Using Hmmer software (version 3.0) with default parameters, *L. indica* protein sequences were analyzed. Furthermore, we verified the structural domains of the MADS family using the online portals SMART (http://smart.embl-heidelberg.de/, accessed on 22 October 2023) and NCBI CDsearch (https://www.ncbi.nlm.nih.gov/Structure/cdd/wrpsb.cgi, accessed on 22 October 2023). Genes with incomplete predictions were excluded from the analysis. The distribution of *LiMADS* genes across chromosomes was illustrated using the “Gene Location” feature in TBtools. The key properties of the LiMADS, including the theoretical isoelectric point (pI), molecular weight (MW), instability index, and hydrophobicity index, were estimated using Expasy (https://web.expasy.org/protparam/, accessed on 22 October 2023).

### 4.2. Phylogenetic Analysis of LiMADS

The protein sequences of *Arabidopsis* MADS were obtained from the Plant Transcription Factor Database (http://planttfdb.gao-lab.org/index.php/, accessed on 22 October 2023). Subsequently, *L. indica* and *Arabidopsis* MADS protein sequences were analyzed using MEGA v11 software and the NJ clustering method with default parameters. Finally, the resulting phylogenetic tree was visualized using iTOL v6 (https://itol.embl.de/, accessed on 25 January 2024).

### 4.3. Chromosome Location and Synteny Analysis of the MADS-Box Gene Family

The MADS-box gene family was located on the chromosome using Gene Location Visualize in TBtools [58], and the genes were named according to their respective positions on the chromosome.

The analysis of homology between MADS *L. indica* and *Arabidopsis* was conducted using the “MCScanX” tool from TBtools. Subsequently, homologous *L. indica* genes were analyzed using the “Advanced Circos” tool. The duplicated gene pairs were identified, and the identity substitution rate (Ks) and non-identity substitution rate (Ka) of these pairs were calculated using the “Simple Ka/Ks calculator” tool. The time of differentiation for the duplicate gene pairs was calculated using the *Arabidopsis* molecular clock (λ) at a rate of 1.5 × 10^−8^ substitutions per site per year, as T = Ks/2λ.

### 4.4. LiMADS Gene Structure Analysis and Cloning

The coding sequence and genome annotation information for the *LiMADS* gene family were derived from the *L. indica* genome database. The MEME v5.5.2 (https://meme-suite.org/meme/tools/meme, accessed on 5 November 2023) online tool was employed to predict the conserved motifs of *LiMADS*, with the motif count set to 10 and all other parameters set to their default settings. The motif data obtained from MEME, along with the genomic GFF data from *L. indica*, were visualized using the “Gene Structure View” feature in TBTools.

The method of gene cloning was to design primers on the CDS sequence of *L. indica* (Appendix A), purify the product after PCR amplification, and determine the sequence size by agarose gel electrophoresis. It was then linked to the pTOPO-Blunt cloning vector (Aidlab Biotechnologies Co., Ltd. Beijing, China), transformed into DH5α, and finally sent to the BGI Genomics company for sequencing. We then uploaded the correct gene sequences to NCBI and obtained the following accession numbers: LiMADS81-Hong Ye (PP388995), LiMADS81-Xiang Yun (PP388996), LiMADS37-Hong Ye (PP388997), and LiMADS37-Xiang Yun (PP388998).

### 4.5. Prediction of Cis-Regulatory Elements in the Promoter Region

The prediction of cis-acting elements for a 2000 bp sequence located upstream of the LiMADS start site was performed using the PlantCARE (http://bioinformatics.psb.ugent.be/webtools/plantcare/html/, accessed on 23 October 2023) and New PLACE (https://www.dna.affrc.go.jp/PLACE/, accessed on 25 January 2024) online software. The resulting data was subsequently visualized through the TBtools v2.052 “Basic Biosequence View” visualization tool.

### 4.6. Analysis of LiMADS Protein-Protein Interaction Network

To investigate protein interactions, the protein sequences derived from the *L. indica* genome were submitted to the STRING database v12.0 (https://string-db.org/, accessed on 3 November 2023), which led to the analysis of the relationships between LiMADS proteins using *Arabidopsis* as a reference, and the results were visualized via the Cytoscape v3.10.1 (https://cytoscape.org/, accessed on 3 November 2023).

### 4.7. Expression Analysis of MADS Gene at Different Stages of L. indica Flower Development

To investigate the expression level of the MADS-box gene during flower development of *L. indica*, we extracted total RNA from ‘Xiang Yun’ and ‘Hong Ye’ flowers at two stages: the bud stage (S3) and the mature flowering stage (S5). Each sample was replicated three times. Novogene Co., Ltd. (Beijing, China) performed transcriptome sequencing. We uploaded the transcriptome data to the NCBI and obtained a login number, PRJNA1071351. Fragments per kilobase per million from mapped readings (FPKM) were used to compare the expression levels of *LiMADS* during two stages of flower development. A cluster heatmap was created using the cluster heatmap tool of TBtools to visually analyze gene expression levels.

We further determined the expression of ‘Xiang Yun’ and ‘Hong Ye’ flower genes at S3 and S5 stages using fluorescence quantitative PCR (Applied Biosystems QuantStudio 6 Flex). ‘Hong Ye’ was used as a control, and the 18S gene was used as a reference gene. Each sample was repeated three times. The reaction system and amplification procedures were set up according to the indicated protocols using standard procedures, and the data were analyzed using the 2^−ΔΔCt^ method. Primers used for vector construction are listed in Appendix A.

### 4.8. Analysis of Hormone Levels at Different Stages of L. indica Flower Development

Flower buds or flowers were collected from ‘Xiang Yun’ and ‘Hong Ye’ grown on the Experimental Forest Farm of the Hunan Academy of Forestry. They were picked at 72 h (S1), 48 h (S2), 24 h (S3), 1 h (S4), and 0 h (S5) before flowers opened. Sam The samples were cleaned with deionized water, frozen in liquid nitrogen, and stored at −80 °C. Fifteen plant hormones (ABA, MeSA, MeJA, GA_3_, GA_4_, JA, IAA, IBA, c-Zeatin, TZR, iP, and iPR, sourced from Sigma, Shanghai, China; SA, GA_1_, and GA_7_, purchased from TRC, Toronto, Canada) were subjected to high-performance liquid chromatography (HPLC). Freeze-dried flower bud samples (0.5 g) were mixed with an acetonitrile volume of 10 times and left to extract overnight at 4 °C. The mixture was then centrifuged, and acetonitrile was added to the supernatant. Centrifugation was performed at 10,000× *g* for 5 min after adding 40 mg of C18 filler to the supernatant, followed by drying with nitrogen and redissolving in methanol. Subsequently, an organic phase filter membrane with a pore size of 0.22 μm and filter size of 200 μ was employed, and the resulting mixture was stored at −20 °C. The SCIEX-Qtrap6500 instrument (AB SCIEX, Redwood City, CA, USA)was utilized for the analysis of phytohormones, with each analyte identified and quantified using the multiple reaction monitoring (MRM) mode. Three independent replicates were used for each phytohormone extraction.

### 4.9. Data Processing

All experimental data were repeatedly measured three times, subsequently averaged, and presented as mean ± standard deviation (SD). To perform a two-way ANOVA multi-factor analysis of variance and generate bar charts, the software program GraphPad Prism 10 was utilized.

## 5. Conclusions

This study identified a total of 82 *LiMADS* genes in *L. indica*, which were classified into five subfamilies. Among them, the *MIKC^C^* subfamily comprises 60% of the genes. The *MIKC^C^* genes are crucial for regulating floral organ development. Analyzing upstream sequences of *LiMADS* indicated cis-acting elements associated with hormonal responses, specifically the methyl JA response elements and gibberellin response elements, occurred in the promoter regions. Endogenous hormones, particularly JAs, IPR, TZR, Zeatin, and Gas, showed dynamic changes during the course of flower development. Correlation analysis showed that JA, IAA, and GAs were positively correlated with most *MIKC^C^* genes. A working model for the development of *L. indica* ‘Xiang Yun’ was proposed. The hormone response pathway may affect the MADS gene family related to flower development and cause its low expression, resulting in abnormal development of stamen and embryo sac, and ultimately affect the outcome process of ‘Xiangyun’. It is anticipated that the findings from this study could serve as a valuable reference for investigating the function of the MADS-box gene family in *L. indica* and mechanisms underlying the sterility of ‘Xiang Yun’.

## Figures and Tables

**Figure 1 plants-13-00709-f001:**
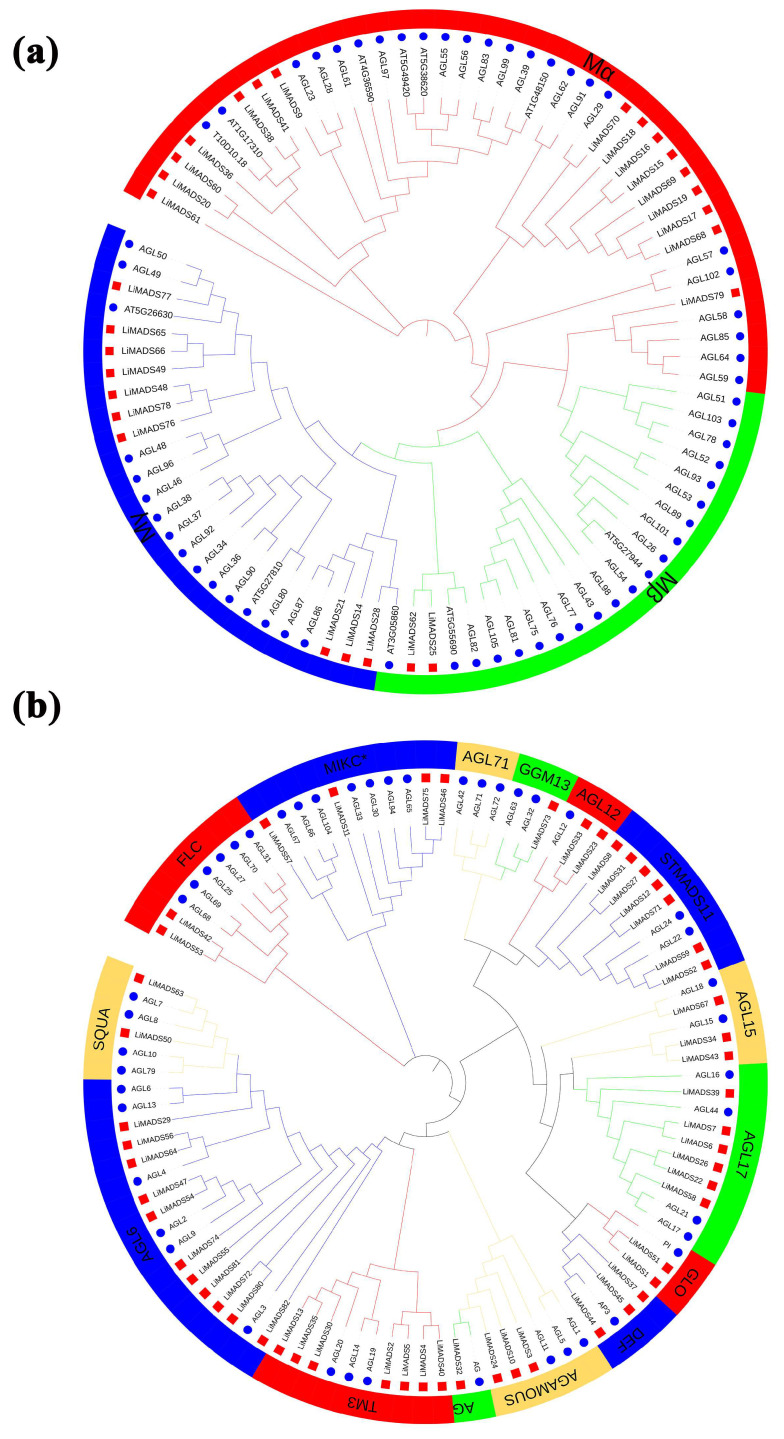
Phylogenetic relationships of MADS-box genes in *L. indica*. The MADS-box gene in *Arabidopsis* is indicated by a blue circle, whereas the MADS box gene in *L. indica* is indicated by a red square. Subfamily names are shown in different colors. (**a**) Type I genes are classified as Mα, Mβ, and Mγ groups; and (**b**) Type II genes are divided into 14 subfamilies.

**Figure 2 plants-13-00709-f002:**
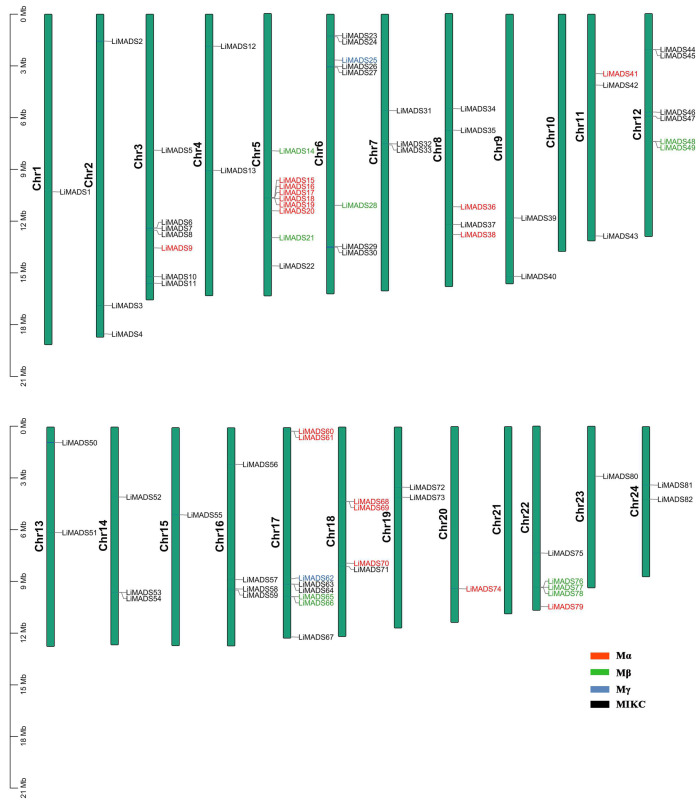
*LiMADS* chromosome distribution in *L. indica*. The chromosome number is located on the left of the chromosome.

**Figure 3 plants-13-00709-f003:**
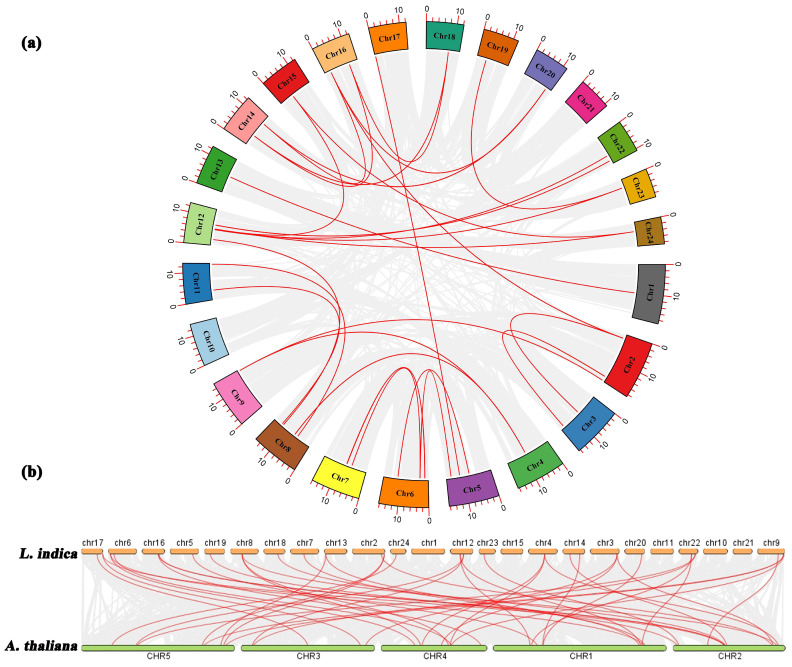
Homology analysis of the *MADS*. (**a**) Collinearity analysis of the *MADS* gene family in *L. indica*, and the red line represents fragment replication within the gene family. (**b**) Co-linear analysis of *MADS* genes in *L. indica* and *Arabidopsis*. Li represents the *L. indica* genome. At represents the *Arabidopsis* genome, numbers represent chromosome numbers, gray lines represent collinear relationships between different genomes, and red lines represent collinear relationships between *MADS* genes.

**Figure 4 plants-13-00709-f004:**
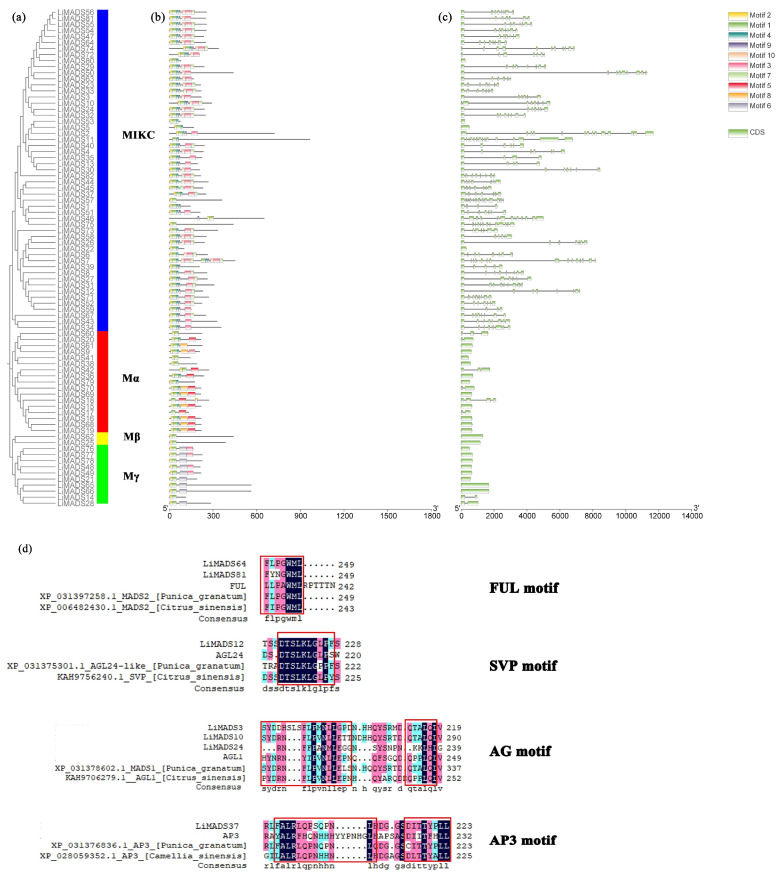
Structural and conserved motif analysis of the *LiMADS* gene in *L. indica*. (**a**) Phylogenetic classification of LiMADS proteins; (**b**) Conservative motif analysis, with different patterns highlighted by colored boxes, the length of which is related to the length of the basic sequence; (**c**) Coding sequence of the *LiMADS* gene; (**d**) C-terminal domain differences between *L. indica* and different species. The red box indicates the motif’s position.

**Figure 5 plants-13-00709-f005:**
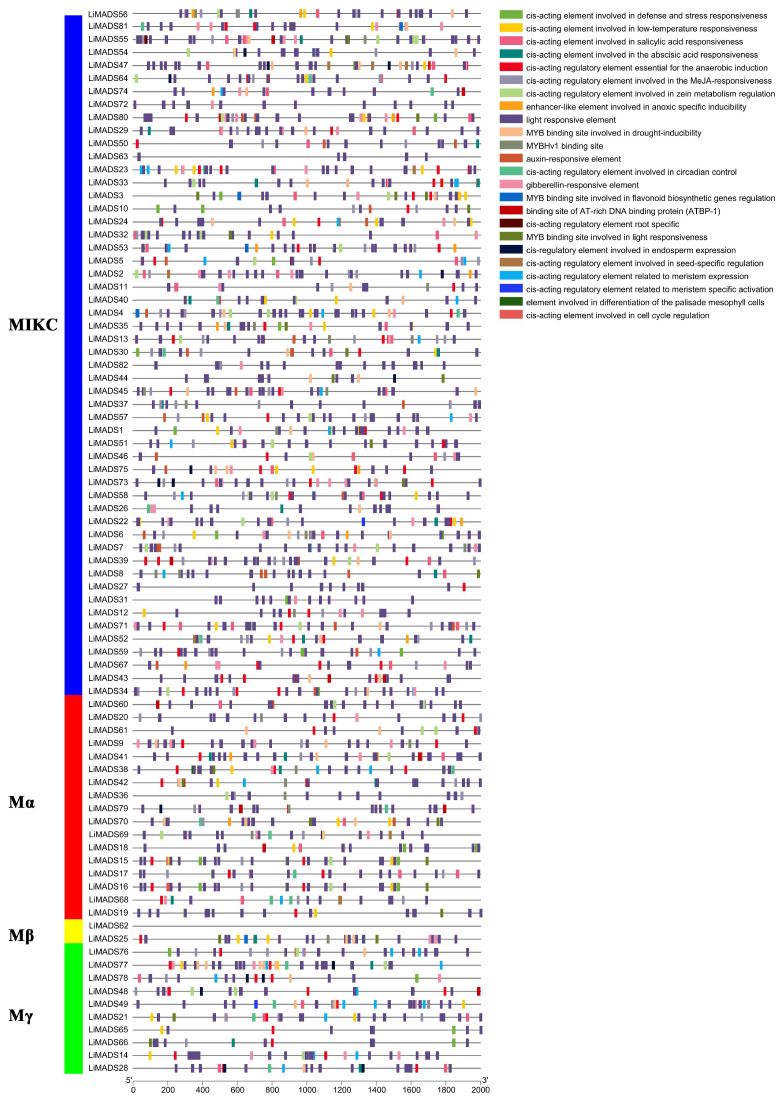
Analysis of cis-acting elements in the promoter region of the LiMADS family. Different cis acting elements are displayed in different colors.

**Figure 6 plants-13-00709-f006:**
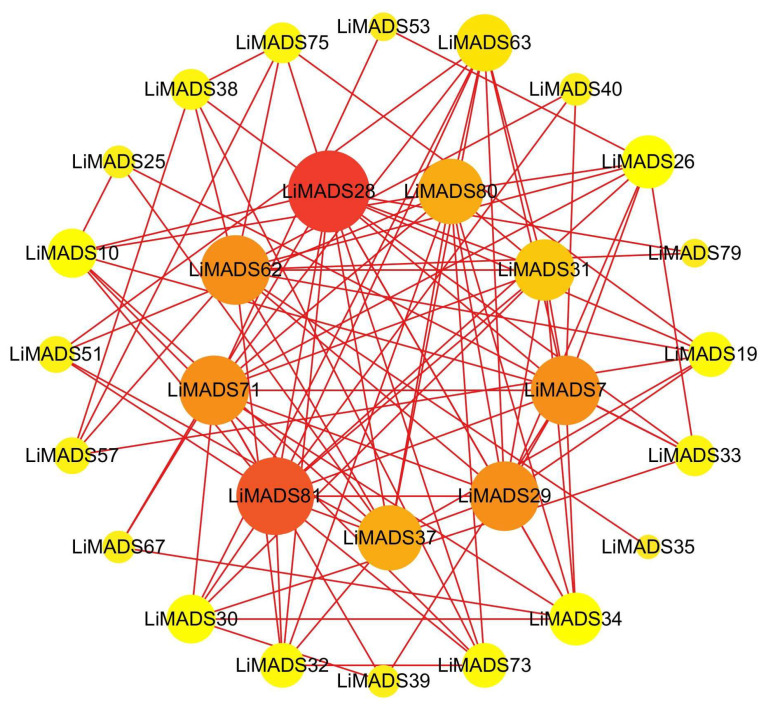
Protein–protein interaction network of the *MADS* gene family in *L. indica*. The depth of the color in the figure and the size of the circle indicate the magnitude of interactions, with red indicating a large degree value and yellow indicating a small degree value.

**Figure 7 plants-13-00709-f007:**
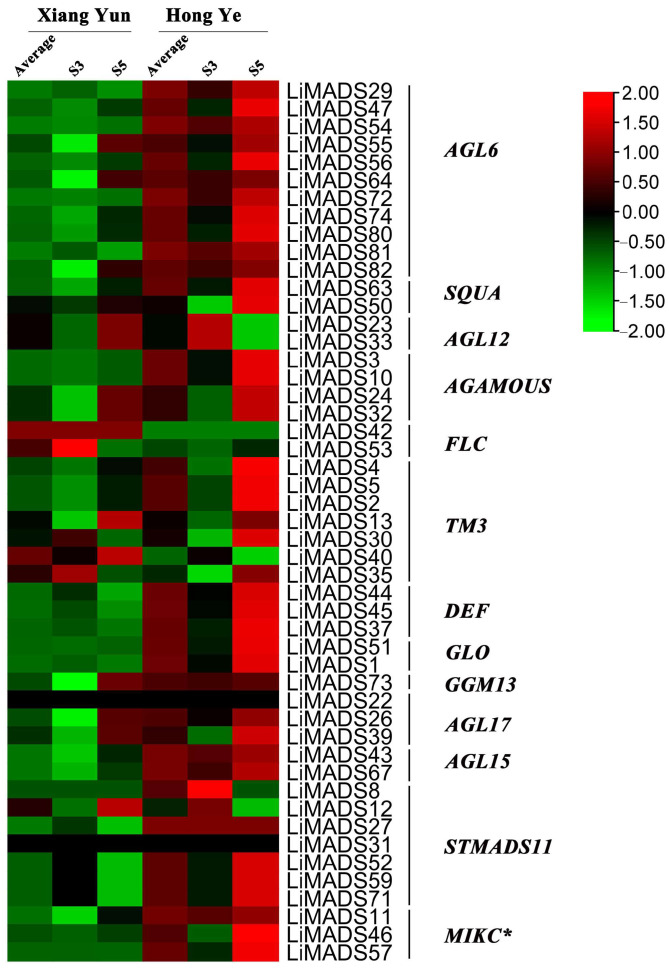
Relative expression of *LiMADS* genes. All genes are classified into different subfamilies, and the color scale indicates relative expression levels from high (red) to low (green).

**Figure 8 plants-13-00709-f008:**
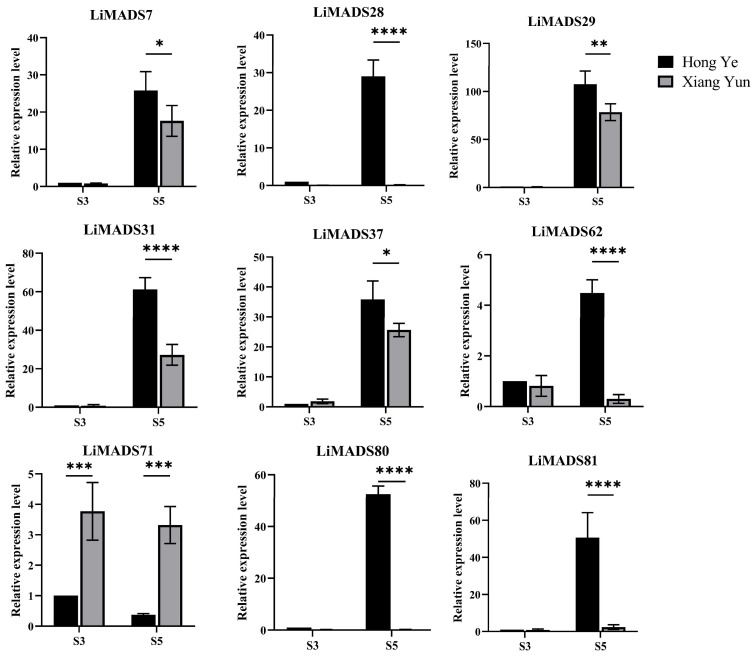
Expression levels of different *MADS* genes in *L. indica* at different flower development stages. The expression analysis was normalized using *Li18S* gene as internal reference gene. Values are means of three independent repeats, and error bars are standard deviations, where *, **, ***, and **** represent significant differences at *p* < 0.05, *p* < 0.01, *p* < 0.001, and *p* < 0.0001 levels based on two-way ANOVA test.

**Figure 9 plants-13-00709-f009:**
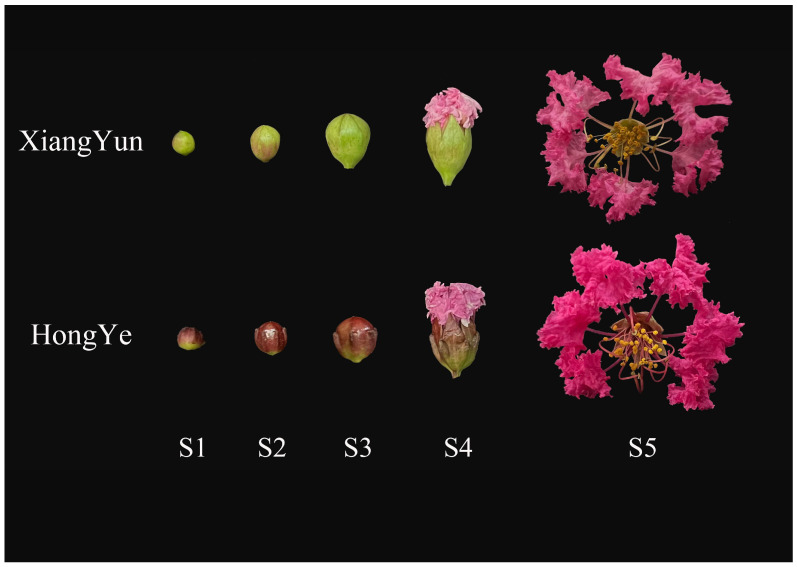
Flower development of *L. indica* ‘Xiang Yun’ and ‘Hong Ye’ during different periods. S1–S5 is the process of flower maturation.

**Figure 10 plants-13-00709-f010:**
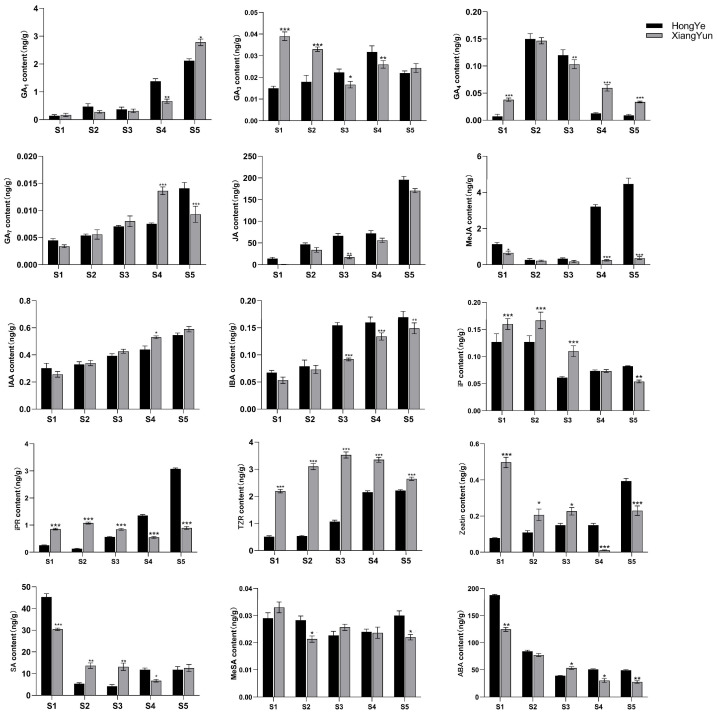
Levels of 15 hormones at different flower development stages of *L. indica*. All data were derived from at least three biological replicates and are expressed as the mean ± standard deviation. (*: *p* < 0.05, **: *p* < 0.01, ***: *p* < 0.001, two-way ANOVA test).

**Figure 11 plants-13-00709-f011:**
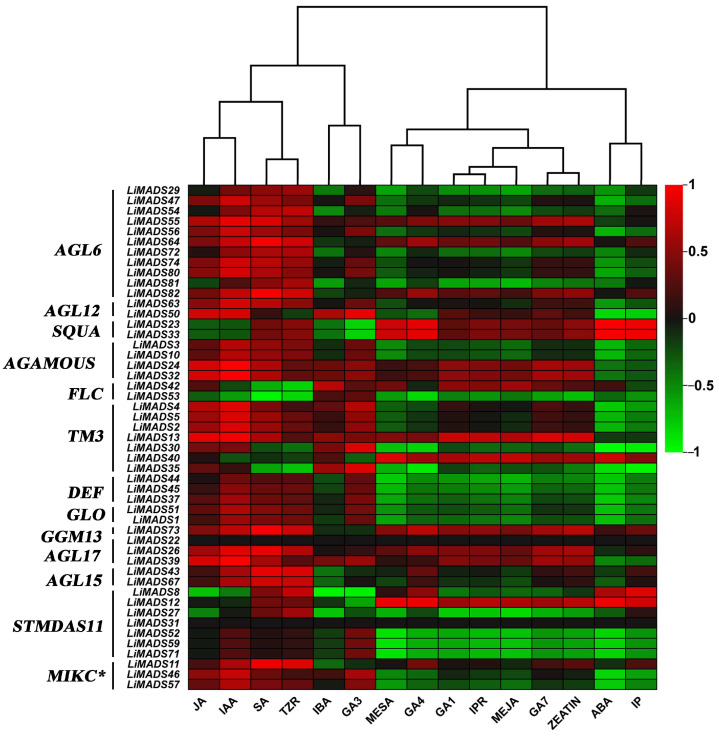
The correlation analysis between different hormones and MADS type II gene. All genes were grouped into different subfamilies, with color scales indicating correlation coefficients from high (red) to low (green).

**Figure 12 plants-13-00709-f012:**
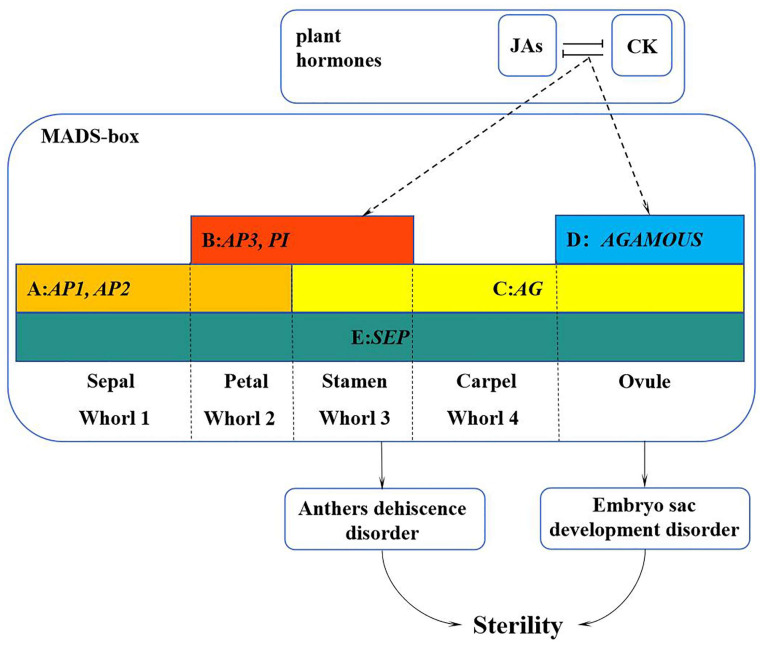
A working model of the MADS-box family genes in regulation of flower development of *L. indica* ‘Xiang Yun’. The sterility of ‘Xiang Yun’ was characterized by pollen inactivity, another dehiscent disorder, and an abnormal tapetum layer leading to embryo sac abortion. The dashed line represents the procedure proposed based on this study.

**Table 1 plants-13-00709-t001:** Molecular weights and physicochemical properties of 82 LiMADS proteins in *L. indica*.

Sequence ID	Chromosome	Number of Amino Acid	Molecular Weight	Theoretical PI	Subfamily	Type	Subcellular Localization
LiMADS1	Chr1	143	16,347.72	9.30	MIKC	Type II	Nucleus
LiMADS2	Chr2	720	81,541.55	6.57	MIKC	Type II	Nucleus
LiMADS3	Chr2	219	25,363.98	9.42	MIKC	Type II	Nucleus
LiMADS4	Chr2	233	26,849.61	9.49	MIKC	Type II	Nucleus
LiMADS5	Chr3	166	18,608.16	9.77	MIKC	Type I	Nucleus
LiMADS6	Chr3	262	29,458.14	9.45	MIKC	Type II	Nucleus
LiMADS7	Chr3	450	51,212.25	7.11	MIKC	Type II	Nucleus
LiMADS8	Chr3	257	28,811.55	9.66	MIKC	Type II	Nucleus
LiMADS9	Chr3	207	22,949.55	9.40	Mα	Type I	Nucleus
LiMADS10	Chr3	290	33,123.44	9.11	MIKC	Type II	Nucleus
LiMADS11	Chr3	964	107,927.12	8.65	MIKC	Type I	Nucleus
LiMADS12	Chr4	229	25,466.16	9.56	MIKC	Type II	Nucleus
LiMADS13	Chr4	192	21,483.46	9.36	MIKC	Type II	Nucleus
LiMADS14	Chr5	112	12,247.65	5.27	Mγ	Type I	Nucleus
LiMADS15	Chr5	217	24,350.51	5.81	Mα	Type I	Nucleus
LiMADS16	Chr5	217	24,292.47	5.81	Mα	Type I	Nucleus
LiMADS17	Chr5	133	15,343.77	8.63	Mα	Type I	Nucleus
LiMADS18	Chr5	272	30,940.98	8.05	Mα	Type I	Nucleus
LiMADS19	Chr5	217	24,442.62	6.02	Mα	Type I	Nucleus
LiMADS20	Chr5	215	24,176.95	8.46	Mα	Type I	Nucleus
LiMADS21	Chr5	188	21,283.61	4.54	Mγ	Type I	Nucleus
LiMADS22	Chr5	100	11,055.98	9.57	MIKC	Type I	Nucleus
LiMADS23	Chr6	214	23,904.68	8.95	MIKC	Type II	Nucleus
LiMADS24	Chr6	239	27,585.52	9.30	MIKC	Type II	Nucleus
LiMADS25	Chr6	385	42,403.25	7.69	Mβ	Type I	Nucleus
LiMADS26	Chr6	242	27,877.93	9.44	MIKC	Type I	Nucleus
LiMADS27	Chr6	261	29,225.49	5.82	MIKC	Type II	Nucleus
LiMADS28	Chr6	282	31,904.09	9.51	Mγ	Type II	Nucleus
LiMADS29	Chr6	238	27,131.80	9.04	MIKC	Type II	Nucleus
LiMADS30	Chr6	208	23,761.39	8.49	MIKC	Type II	Nucleus
LiMADS31	Chr7	307	34,400.55	6.36	MIKC	Type II	Nucleus
LiMADS32	Chr7	247	28,520.48	8.77	MIKC	Type II	Nucleus
LiMADS33	Chr7	217	23,984.56	7.74	MIKC	Type II	Nucleus
LiMADS34	Chr8	354	40,377.15	7.63	MIKC	Type II	Nucleus
LiMADS35	Chr8	224	25,635.33	8.89	MIKC	Type II	Nucleus
LiMADS36	Chr8	235	25,934.06	5.89	Mα	Type I	Nucleus
LiMADS37	Chr8	251	28,951.37	6.04	MIKC	Type II	Nucleus
LiMADS38	Chr8	189	21,173.81	5.73	Mα	Type I	Nucleus
LiMADS39	Chr9	206	22,516.08	9.13	MIKC	Type I	Nucleus
LiMADS40	Chr9	239	27,232.87	9.49	MIKC	Type II	Nucleus
LiMADS41	Chr11	143	15,956.17	10.31	Mα	Type I	Nucleus
LiMADS42	Chr11	271	29,674.12	5.30	MIKC	Type I	Nucleus
LiMADS43	Chr11	326	35,876.75	6.05	MIKC	Type II	Nucleus
LiMADS44	Chr12	266	30,702.20	8.73	MIKC	Type II	Nucleus
LiMADS45	Chr12	230	26,301.00	8.86	MIKC	Type II	Nucleus
LiMADS46	Chr12	650	73,648.67	6.78	MIKC	Type I	Nucleus
LiMADS47	Chr12	236	27,188.86	6.46	MIKC	Type II	Nucleus
LiMADS48	Chr12	212	23,842.25	5.88	Mγ	Type I	Nucleus
LiMADS49	Chr12	216	24,415.90	5.90	Mγ	Type I	Nucleus
LiMADS50	Chr13	1784	199,613.24	9.01	MIKC	Type II	Nucleus
LiMADS51	Chr13	210	24,598.99	9.03	MIKC	Type II	Nucleus
LiMADS52	Chr14	223	25,281.74	5.95	MIKC	Type II	Nucleus
LiMADS53	Chr14	75	8748.21	9.93	MIKC	Type I	Nucleus
LiMADS54	Chr14	252	28,587.54	8.24	MIKC	Type II	Nucleus
LiMADS55	Chr15	255	28,989.91	7.59	MIKC	Type II	Nucleus
LiMADS56	Chr16	254	29,244.13	8.29	MIKC	Type II	Nucleus
LiMADS57	Chr16	361	40,440.63	5.18	MIKC	Type I	Nucleus
LiMADS58	Chr16	253	29,160.32	8.95	MIKC	Type II	Nucleus
LiMADS59	Chr16	150	17,238.80	9.14	MIKC	Type II	Nucleus
LiMADS60	Chr17	224	25,120.97	8.67	Mα	Type I	Nucleus
LiMADS61	Chr17	225	24,916.43	7.67	Mα	Type I	Nucleus
LiMADS62	Chr17	440	49,045.53	7.14	Mβ	Type I	Nucleus
LiMADS63	Chr17	166	19,540.50	9.86	MIKC	Type II	Nucleus
LiMADS64	Chr17	249	28,410.38	8.88	MIKC	Type II	Nucleus
LiMADS65	Chr17	560	61,472.15	6.55	Mγ	Type I	Nucleus
LiMADS66	Chr17	560	61,673.41	7.11	Mγ	Type I	Nucleus
LiMADS67	Chr17	249	27,930.73	6.56	MIKC	Type II	Nucleus
LiMADS68	Chr18	218	24,644.92	5.92	Mα	Type I	Nucleus
LiMADS69	Chr18	215	24,199.27	6.01	Mα	Type I	Nucleus
LiMADS70	Chr18	216	24,547.74	6.78	Mα	Type I	Nucleus
LiMADS71	Chr18	269	30,168.43	7.70	MIKC	Type II	Nucleus
LiMADS72	Chr19	208	23,887.52	9.40	MIKC	Type II	Nucleus
LiMADS73	Chr19	331	38,325.46	8.77	MIKC	Type II	Nucleus
LiMADS74	Chr20	337	37,643.01	8.73	MIKC	Type II	Nucleus
LiMADS75	Chr22	439	49,794.41	7.32	MIKC	Type I	Nucleus
LiMADS76	Chr22	164	18,941.46	9.66	Mγ	Type I	Nucleus
LiMADS77	Chr22	225	25,846.28	8.82	Mγ	Type I	Nucleus
LiMADS78	Chr22	225	25,501.79	6.45	Mγ	Type I	Nucleus
LiMADS79	Chr22	173	19,408.45	10.04	Mα	Type I	Nucleus
LiMADS80	Chr23	79	8773.17	9.84	MIKC	Type I	Nucleus
LiMADS81	Chr24	249	28,425.17	7.68	MIKC	Type II	Nucleus
LiMADS82	Chr24	214	24,851.74	9.77	MIKC	Type II	Nucleus

## Data Availability

The data presented in this study are available in Appendix A.

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
