# Peer review of "MADS-Box Family Genes in *Lagerstroemia indica* and Their Involvement in Flower Development"

_plants, 2024, doi:10.3390/plants13050709_

Round 1
Reviewer 1 Report
Comments and Suggestions for Authors
General comments :
The identification of MADS-box family genes in Lagerstroemia indica was reported by the authors. Additionally, the authors presented findings from an RNAseq study and hormone analysis conducted during fruit development. The manuscript is unique and comprises intriguing information.
However, I have general comments regarding the manuscript.
The manuscript compares the promotors of all MADS-box gene families in this species and identifies the putative cis-element. However, the authors did not discuss the exact difference between promoters. Also, is there any conserved cis-acting of this gene family promoters? Especially between subgroups? Please discuss this result.
Furthermore, with regard to the comparisons of promoter structures, are there any 5' UTR introns within the promoters, information about the transcription start site (TSS), and so on? This is crucial for a comprehensive understanding of the variations in gene expression levels and the specificity of gene expression, particularly in terms of tissue-specific expression. Such data can be extracted from RNAseq data. Currently, the authors have only reported on putative cis-acting elements. I recommend that the authors incorporate this additional information into their manuscript.
The experiment used in this study only used transcriptomic RNA-Seq. There is no RT-qPCR data for gene expression validation. I would like to recommend this to support the differential gene expression gene reported in the manuscript.
There is no supplementary in the documents. I suggest to include this in the revised version.
Also, please modify the material and methods section. The sample used in the study and design experiment is not clearly described.
A lot of statement including the conclusion is not supported by the data. For example, please provide a statistical correlation between hormone and RNAseq data that supports the statement or it should be conducting the experiment on it.
I would like to suggest that the authors conduct the alignment of the MADS-box gene between the mutant and wild genotypes. Is there any polymorphism between the genotypes?
I found many research questions that are still unknown regarding these MADS-box transcription factors. I would recommend including the limitations or future research steps in the manuscript.
Specific :
Line 150. “Two different floral developmental stages”. Which stages do you mean? you can refer to the development stages in Figure 8. Please include the variety you used in this study. Please revise this and make it more clear.
Line 152. Please include L. indica transcriptome sequencing data public repository accession number (eg. NCBI or else) because it is necessary. Even though the original paper is not yet published.
Line 192. Table 1. Did the authors check all of the gene's localization? I would suggest to include this information.
Line 195 and 197. L. indica and Arabidopsis thaliana should be in italics. Please revise this throughout the manuscript.
Line 231. “Analysis of cis-acting elements”. Please revise cis-acting elements into putative cis-acting elements. Since the authors only predicted the cis-acting motif/element using bioinformatics not from experimental data. Please revise this throughout the manuscript.
Line 246. Figure 2 is not really informative, because the colors of putative cis-acting cannot be clearly differentiated from each other. Using a number or code would be much more helpful for the reader. I would suggest to change it.
Line 276. Figure 3. Even though there is no MADS gene in chr 10 and 21. However, you can still include these chromosomes also in the figure. To avoid misunderstanding that this species missing these chromosomes.
Line 289. You find a conserved motif in this MADS-box gene family of this species. Please discuss your comparison with other species that are known previously. Did you find a new motif?
Line 333. Please include information regarding the color meaning (For example, blue, green, and red group).
Line 318. There is no discussion regarding the MADS-box protein-protein interaction result. Please provide this in the discussion section. For instance, what is the implication of the protein interaction to the flower development? is it as expected compared to other species ?. etc. In addition, the authors can include references and experiment data so that the MADS-box can interact with each other.
Line 360. Figure 7. Please include the sample name in the figures. For example, replicate 1,2,3. Also, do you use normalization to create the heatmap?
Line 373. Figure 8. What is the age of the flower S1-S5 stage? Days before anthesis or Days after anthesis? This is not mentioned in the material and methods also.
Line 399. Figure 9. What is the result of two-way ANOVA? Please explain clearly which of two factors that really influences the difference between two cultivars.
Line 431-432. Is there any unique cis-acting that is not found in other species?
What is the difference between promoters? is there any specific cis-acting between sub-groups? This is related to general comments.
Line 482. This summary is not supported by the data. Please provide a statistical correlation between hormone and RNAseq data ?
Line 488. Figure 10. Please elaborate on the sterility phenomenon in Xiang-Yun. The problem should be included in the introduction that the non-fruiting is caused by either anthers or embryo disorder? or both have a problem in this variety.
Line 498. “Plant hormones responded to the decreased expression of class B, C, and D genes through antagonism with CK”. This conclusion is not supported by the data. Please provide supporting data either by statistical analysis or experiment.
Line 498. Please keep in mind that you measure the hormone IPR, TZR, Zeatin, etc, this is a specific type of hormone, not a cytokinin hormone in general. Therefore, I would suggest carefully taking the conclusion. Only mention what you measured.
Comments on the Quality of English LanguageMinor editing of English language required
Author Response
Please see attached file "Responses to Reviewer 1"

Reviewer 2 Report
Comments and Suggestions for Authors
The work describes association of MADS-box transcription factors in the ornamental Lagerstroemia indica ´Xiang Yun´, and its behavior associated to a non-fruiting trait. The study includes hormone characterization and flower development.
Gene identification. Considering that the work includes “big data analysis” to define MADs-box genes in the species, authors should prioritize the base information to be delivered in the main text section. For instance, in the Gene Identification subsection, authors should invest efforts in summarizing concepts in a single paragraph depicting the hot spots derived from the first findings. This means, current Table 1 should be sent to supplementary material, and then to include a single paragraph summarizing that information instead (i.e., the number of nucleotides of all the 82 genes coding domain sequence varied from XXX bp to YYY bp, the number of amino acids encoded 82 MADS-box proteins varied from ZZZ aa to RRR aa, and the predicted relative molecular mass ranged from AAA to BBB kDa, with protein isoelectric point in the range of QQQ to WWW.
Figure 1 should be improved. Please, make efforts clarifying the represented/identified groups.
Considering that authors will give a chromosome distribution and homology analysis, I´m not sure if promoter analyses (cis-acting elements) should be (at this rate) in the following section of the document. Can authors explain why they proposed this organization (i.e., cis-elements before chromosome dist. and homology analysis)?
In fact, are these elements (chromosome distribution and homology analysis) needed in these sections of the manuscript?
In the same way, phylogenetic comparisons including exon-intron gene structures and conserved motifs in the gene bodies, should at least, keep the type I and II family indication (Figure 5). This will bring clarity to the reader.
Then, questions arise from the next components in the paper: how much needed is to show cis-acting elements in promoter areas for all the proposed (found) genes. Considering the focus on flower development and differential hormone profiling between ´Xiang Yun´ and ´Hong Ye´, it is possible to create a selection showing only genes that will deserve observation for those next points (as in Figure 7)? Which response elements (in particular/selected members of the MADS family) are relevant considering functional characterizations (flowering and hormone content)?
In this line, protein-protein networks at level of point 3.6 in Results seem too speculative. Whereas the next paragraphs were focused on differential expression either by flower development (section 3.7) and hormone profiles (section 3.8), the current Figure 6 is describing models with no support at section 3.6. According to this, Figure 6 can be moved as part of Figure 10, under Discussion. Is Figure 6 needed?
Missed information in Figure 7, no stages included. In this context, in the Materials and Methods section, authors indicate “two different developmental stages” (2.6). I did not understand why Figure 7 (heat map) is depicting three lanes for each variety. Regardless this point, methods for these experiments (section 2.6) must be established.
Also, Methods section regarding hormone determinations must be re-built indicating sampling procedures, conditions, and developmental stage(s) considered.
No supplementary materials were found in the current submission.
Comments on the Quality of English LanguageSome minor-moderate checking must be carried out on the new version of this manuscript.
Author Response
Please see attached file "Responses to reviewer 2"

Round 2
Reviewer 1 Report
Comments and Suggestions for Authors
The revised manuscript is significantly improved. However, I still found some minor mistakes. Please revise this before publication.
Figure 2. Please revise the figure with the new version as in the response documents.
Line 486. "15 hormones" should be "16 hormones".
Line 496. Figure 11 legends. "color scales indicating relative expression levels from high (red) to low (green)". Is this correct or is it supposed to be a correlation value? Please confirm again.
Line 148. please revise "L.indica" in italic , also please check again throughout the manuscript.
I would like to recommend including the NCBI accession number of the cloned LiMADS sequence (wild & mutant). This information will be useful for the reader.
Again, thank you for considering all of my suggestions and comments.
Author Response
Thank you so much again for reviewing our manuscript. Our responses to your concerns are included in the attached file.

Reviewer 2 Report
Comments and Suggestions for Authors
In the present version (R1), the manuscript has incorporated (and expanded) all my previous observations. Authors have dedicated good efforts clarifying the work as much as possible, and additionally, they have incorporated new data. This reviewer thanks to the authors their dedication and consideration in improving the work. Indeed, as authors indicated in the response, they followed in some sections, the same structure found in similar works, which is solid.
Authors have incorporated in R1 the correlation between hormone concentrations and gene expression levels in the transcriptome. This component in the work consolidates previous sections in Results, supporting the hormone determinations. Although these inclusions have extended the manuscript. Regardless this point, the findings added by these elements will certainly improve the impact of the work.
In this regard, Discussions were leveled according to these improvements.
Captions are adequate to figures.
Finally, Supplementary materials allow for a deeper study of the work.
Other.
Lines 488-492 (Figure 11). Please, indicate (remark) the genotype (variety; X.Y.??) for which hormone-transcript associations were analyzed and the developmental stage(s) covered by those determination(s). I didn´t find them in the results´ paragraph neither in the caption for Figure 11. Please, check carefully this issue (these details) considering the current status of the paper.
Lines 532-539. Please, consider that proposed prot-prot interactions are deduced from bioinformatic analyses.
Throughout the text. Please, check species´ name nomenclature; complete italics (at first) and italics abbreviated (A. thaliana) after the first mention (for instance lines 385 and 390; 238 and 259, and may be others).
Author Response

(The authors gave the same response as above.)
